# Rheology of Highly Concentrated Suspensions with a Bimodal Size Distribution of Solid Particles for Powder Injection Molding

**DOI:** 10.3390/polym13162709

**Published:** 2021-08-13

**Authors:** Anton V. Mityukov, Vitaly A. Govorov, Alexander Ya. Malkin, Valery G. Kulichikhin

**Affiliations:** A. V. Topchiev Institute of Petrochemical Synthesis, Russian Academy of Sciences, 119919 Moscow, Russia; ant-mityukov@yandex.ru (A.V.M.); vitaly.govorov@yandex.ru (V.A.G.); alex_malkin@mig.phys.msu.ru (A.Y.M.)

**Keywords:** bimodal size distribution, concentrated suspensions, elasto-plasticity, phase separation, powder injection molding, aluminum oxide

## Abstract

Powder injection molding (PIM) is one of the modern and prospective technologies in processing different materials. We proposed to use bimodal compositions of particles for increasing their content in the final products. A set of model suspension of Al with low-molecular-weight poly (ethylene glycol) as a binder based on theoretical arguments concerning the filling capacity of bimodal suspensions was prepared. Studying the rheological properties of these compositions showed that they demonstrate elasto-viscous behavior with significant plasticity that is favorable for the technological process. Using compositions with bimodal distributions allows for increasing the content of the solid phase up to 75 vol. % for PIM technology, which is significantly higher than the standard practical limit. This rheological approach developed for model formulations was applied to processing compositions containing aluminum oxide as typical ceramics and polyolefines as a binder widely used in technological practice. The obtained sintered ceramic samples have quite acceptable mechanical properties of the usual corundum articles.

## 1. Introduction

The powder injection molding (PIM) technology became rather popular in the last few years, due to its possibility to utilize waste and produce rather delicate articles from metals and ceramics. This technology is a combination of two areas of manufacturing: polymer injection molding and powder technology. There are four basic stages: preparing a feedstock, molding, debinding and sintering. Polymer binder and metal or ceramic powder are two main components of the feedstock. Polymers, in combination with some additives, provide a possibility of processing the various parts from feedstocks. However, all components except the powder should be removed after injection molding in order to obtain a metal or ceramic item. This stage is called debinding. The debinding stage can have different versions such as thermolysis or removing by dissoilution, or catalytic debinding. The final stage is sintering when powder particles are sintered with each other [1,2]. From the fundamental point of view, this technology is based on the rheology of highly concentrated suspensions. So, understanding the behavior of such suspensions is necessary for developing this technological process.

Physical, colloidal and rheological properties of suspensions are the topics of the active studies in terms of both academic and applied interest. The majority of systematic investigations were devoted to the rheology of monodisperse solid (non-deformable) particles. Various approaches for characterizing concentration dependence of viscosity were obtained for such suspensions [3,4]. Among other interesting rheological-structure properties of concentrated suspensions, the existence of the maximum volume degree of the filling, *φ_0_* (close to 0.73), which is equivalent to mechanical glass transition, has a principal value [4,5,6,7]. A much higher degree of filling can be reached for “soft” (deformable) particles, since they change their shape under pressure. This is a phenomenon typical for emulsions when the concentration of the disperse phase droplets may even reach 0.98 [8,9].

Another important rheological feature of suspensions is the yield point observed at some threshold concentration *φ****, characterizing a suspension as a visco-plastic medium. The yield point value depends both on the nature of the particle-particle and the particle-dispersion medium interactions. Additionally, this value determines the transition from the solid-like state (gel-like) of material to the flow. Depending on the structure of the suspension, the yield point can be reached even at very low concentrations when supramolecular structures are formed [10]. The concentration interval between *φ**** and *φ_0_* is the domain of possible processing of suspensions in various technological operations, as all such operations are related to the irreversible displacement of material units.

Although the study of monodisperse suspensions looks rather attractive in terms of simplicity of experimental data interpretation, the practical application requires a transition to the area of higher concentrations, since the disperse phase contains those valuable components that are of technological interest.

One of the most obvious approaches to increasing the concentration of dispersion with a volume concentration exceeding *φ_0_* is the transition to the polydispers mixtures. The idea of this approach is demonstrated in Figure 1: the smaller particles fill the free volume between the larger particles, thereby increasing the total volume concentration of the disperse phase.

At the present moment, such highly concentrated suspensions are used in many technologies, including dentistry, cosmetics, chemical power sources, construction, 3D printing, and composite materials production [11,12,13]. Attention should be also paid to the PIM technology in which the achievement of the maximal content of a solid phase is the crucial factor to form high quality and defect-free products [14,15,16].

A theoretical analysis of binary low-concentrated mixtures has led to relations between the viscosity and the ratio of particle diameters [17]. The determination of the largest possible degree of filling depending on the specified parameters is of general value for such systems. This issue has been discussed for emulsions [18,19,20], colloidal liquids [3,21,22] and non-colloidal suspensions [23,24]. In particular, it was shown that there is an optimal value of *λ*—the particle diameters ratio, and *ξ—*the volume content ratio, where the minimum values of filling and viscosity can be achieved [25]. The significant influence of these parameters on the rheological properties of bimodal suspensions was also considered in [26,27,28,29]. Currently, there is a substantial amount of experimental data, theoretical approaches and analytical relationships, which link the relative viscosity of a bimodal non-colloidal suspension to its composition. This is both an empirical equation [30] and a generalized approach [31], based on a huge array of published experimental data [32]. Examples of the correlation between the calculated viscosity of binary dilute and semi-dilute suspensions and the real viscosity of bimodal polystyrene latexes were demonstrated in [33,34,35]. Such correlations for highly filled polystyrene latexes also included the viscosity modeling and calculation of the maximum degree of filling [36]. The use of particles with a diameter ratio of 4.76 and a fine phase volume content in the range of 0.15–0.2 makes it possible to achieve the minimum viscosity values [37]. Besides, it was shown [38] that the bimodality allows for shifting the jamming (i.e., the value of *φ_0_*) that occurs at a larger volume fraction of the solid phase, and therefore leads to a decrease in the viscosity for a given volume fraction of particles.

The paper [39] describes the rheological behavior of bimodal suspensions in the range of the linear viscoelasticity, the concentration dependence of a yield point, as well as the dilatant behavior of a highly filled nano-silica suspension in PEG-400. A model taking into account the effect of particle compaction for the viscosity of high-filled bimodal suspensions was proposed. This model gives results that are in good agreement with experimental data in the range of *λ* values from 1 to 7 [40]. The results of computer simulations, which demonstrated the possibility of increasing the maximum volume fraction of solid spherical particles up to 0.827 at *λ* = 10 and *ξ* = 0.75/0.25 are of particular interest [41]. The simulation also showed a weak relationship between the dense statistical packing and the viscosity of the liquid.

So, as discussed above, as well as in [16,41,42], the use of particles with a sufficiently large ratio of diameters and volume fractions allows us to obtain a suspension with a total solid phase content exceeding 75 vol. %.

This opportunity seems promising in terms of the development of highly concentrated suspensions used in the PIM technology. However, such suspensions are characterized by jamming and the absence of the flow domain that excludes a possibility of processing [4]. Consequently, a concentration of about 65 vol. % is usually considered as the limit in the PIM technology [43]. At the same time, in our earlier publications [44,45] it has been shown that there is a concentration range of plasticity (not flow) where a possibility of irreversible displacement still exists. So, it would be interesting, and possibly useful, to spread this concept on binary suspensions, as well as to try to obtain articles from such suspensions by PIM from as high concentrated suspensions as it can be possible.

This means that the goals of this work are to prepare binary mixtures of the optimal composition that allows for retaining the plasticity up to high concentrations of the filler, to clarify the quantitative characteristics of the rheological properties of such mixtures, to compare their properties with previously studied monodisperse suspensions of similar composition and to form representative articles by the PIM method.

## 2. Materials and Methods

The rheological properties of monodisperse highly concentrated suspensions of the same type like those used in this study were described in detail in our previous publications [44,45]. In this work, samples of bimodal suspensions with compositions that are optimal from the point of view of volume filling were prepared.

Aluminum powders with different average diameters were used for preparing the compositions with the different ratios of diameters: the powder PAD-1 with the average particle size 24 μm, and two types of high-purity aluminum powder, 58 μm and 225 μm. All powders were produced by Rusal, Russia. Then, two model suspensions with *λ* = 2.4 and *λ* = 9.4 were prepared. The shape of powder particles is close to spherical, density is 2.7 g/cm^3^. Low molecular weight polyethylene glycol (MW = 400, viscosity at 25 °C is 0.11 Pa∙s) was used as a continuous medium in model suspensions.

Figure 2 shows the particle size distribution diagrams for the above-mentioned mixtures obtained with an LA-350 particle size analyzer, Germany (a) and an Analysette 22 NanoTech Plus laser particle analyzer, Germany (b).

In the course of the work, suspensions with a total volume concentration of 75 to 85 vol. % were prepared and studied—in other words, concentrations were above the *φ_0_* value for monodisperse suspension. Additionally, the share of smaller particles relative to the total solid phase in suspensions was varied. Suspensions with 75 and 80 vol. % of aluminum powders were prepared with the following ratio of particles of larger and smaller sizes: 80/20, 75/25, 70/30 and 60/40. Suspensions with concentrations of 82.7 and 85 vol. % with a particle ratio of 75/25 were also studied. The concentration of 82.7 vol. % was chosen based on the modeling data in [41]. According to modeling, 82.7 vol. % is the limiting concentration for a bimodal suspension with spherical particles at *λ* = 10, and the fraction of particles of smaller size was 25%. Moreover, a system with a solid phase content of 85 vol. % was prepared to estimate experimentally the filling limit for retaining continuity of suspensions.

Samples of suspensions based on PEG-400 were prepared at room temperature (25 °C), gradually adding the solid phase in the required proportions into the dispersion medium, and then intensively stirring the mixture for 10–15 min.

In addition, we prepared a composition for injection molding based on a bimodal mixture of aluminum powders, in which the matrix is a mixture of food-grade P-2 paraffin (T_m_ = 52–58 °C, ρ = 0.88–0.915 g/cm^3^) and polyethylene LDPE-15803-020 (produced by Gazprom Neftekhim Salavat, Ltd., Russia) with T_m_ = 115 °C, ρ = 0.919 g/cm^3^. Melt Flow Index was 2 g/10 min. Oleic acid used as plasticizer was added to the composition during its preparation.

The molding composition contains 75 vol. % solid phase (*λ* = 9.4), where smaller particles (24 μm) comprise 25% and larger particles (225 μm) comprise 75% of the disperse phase. Binder consists of 24 vol. % of polyethylene and paraffin (1:1), and 1 vol. % of plasticizer. A laboratory rotary mixer HAAKE PolyDrive, Germany was used to prepare the molding composition. The components were loaded into the mixer chamber at a temperature of 150 °C and mixed for 1 h at 30 rpm.

Figure 3 shows micrographs of 80 vol. % suspensions (*λ* = 2.4) with the fraction of the smaller phase of 20 and 40%. Micrographs were obtained using scanning electron microscopy on a MIRA 3 TESCAN device under variable vacuum in the back-scattered electrons (BSE) shooting mode, USA.

The theoretical possibility of using bimodal powder mixtures was applied to the manufacture of aluminum oxide parts. Aluminum oxide powders with an average particle size of 6 and 2 μm produced by LLC Technoceramica (Russia) were used. The protocol for preparing and injection molding of compositions with binary aluminum oxide was similar to the protocol used for model compositions with aluminum.

Rheological studies were carried out on an RS-600 rheometer (ThermoHaake, Karlsruhe, Germany). A plate-plate measuring unit was used, which is a pair of steel grooved discs with a diameter of D = 20 mm. The surfaces have teeth in the form of pyramids of 0.5 mm in height with a square base (1 mm side). The distance between the plates was 1.5 mm.

The experiments were performed in the controlled shear stress mode. The following values of shear stresses were used: 500, 1000, 3000, 5000, 7000 and 10,000 Pa. The stresses were applied for 300 s, after which the loading was ceased and the partial recovery of the deformed sample was observed, also for 300 s.

Thus, the elastic and plastic components of the total deformation were measured as a function of stress. In this case, the upper-stress limit was determined by whether the deformation remains constant, since when this limit was exceeded, breakdown occurs, i.e., we saw either the separation of the sample from the boundary surface or the appearance of a rupture inside the sample [46].

In addition, experiments were carried out to determine the components of complex dynamic modulus in the mode of low amplitude oscillations in the frequency range from 0.1 to 100 Hz. Measurements of the rheological properties in all cases were made at room temperature and repeated three times for each sample.

Injection molding of samples was performed using an IM12 laboratory injection molding machine (Xplore, The Netherlands). This machine has the following technical specification: maximum shot volume 12 cm^3^, programmable cycle run, maximum injector temperature 400 °C. Material is injected into the temperature-controlled mold with a plunger by compressed air. The plunger diameter is 12 mm, and the nozzle diameter is 4 mm. The sample preparation conditions were as follows: melt temperature 150 °C, mold temperature 50 °C, injection pressure 6 atm, holding pressure 6 bar and holding time under pressure 30 s.

Based on these parameters, volumetric flow rate (Q) was calculated:
(1)Q=V/t
(2)$$Q=6\times 10^{-6}/4=1.5\times 10^{-6}m^{3}/s$$


The shear rate on the barrel wall (${\dot{\gamma }}_{b}$) and nozzle wall (${\dot{\gamma }}_{n}$) were calculated as well:
(3)$$\dot{\gamma }=4Q/\pi R^{3}$$
(4)$${\dot{\gamma }}_{b}=4\times 1.5\times 10^{-6}/3.14\times 0.006^{3}=8.8\;s^{-1}$$
(5)$${\dot{\gamma }}_{n}=4\times 1.5\times 10^{-6}/3.14\times 0.002^{3}=3500\;s^{-1}$$


Two sample items were prepared: standard specimens for mechanical testing with cross-sectional dimensions 2 × 4 mm and cylinders with L = D = 15 mm.

## 3. Results and Discussion

### 3.1. Rheology of Compositions

First of all, the studied samples are elastic materials. This is evident from Figure 4 and Figure 5 by the independence of the storage modulus on frequency in a wide range of frequencies measured at the minimum possible deformation amplitude of 0.001 (0.1%), at which reliable stress values can be obtained. Additionally, in all cases G” << G.

The concentration of 82.7 vol. % was the upper limit of filling in our experiments. A composition with a solid phase of 85 vol. % cannot create suspension because the material does not retain continuity. An increase in the concentration of the solid phase leads to a gradual increase in the elastic modulus.

It is interesting to mention that suspensions under study demonstrate the linearity of viscoelastic properties in the wide stress (deformation) range. It is seen from amplitude dependence of the elastic modulus (Figure 6) where the linearity limit takes place at 30% of strain. It means that suspensions retain some free volume at rather high concentration, possibly due to chaotic particles distribution in space. The observed threshold of the linearity is even higher than for model suspension close to 10% [47].

The typical results of experiments performed in the “shear-recovery” mode are shown in Figure 7. If shear stress is applied for suspensions with a solid phase of 75 vol. % and more, constant values of deformations appear faster than 1 s—so, they do not demonstrate viscous flow. After cessation of loading, deformation is instantaneously recovered to some extent, and then remains unchanged for a long time. So, the observed behavior should be treated as the elasto-plasticity, similar to that described for monodisperse suspensions [44]. However, it is important that for this highly concentrated suspension, plasticity (the irreversible component of deformation)—but not the flow—is the dominant part of the total irreversible deformation. Indeed, deformation at the right parts of this Figure having the sense of irreversible part of the full deformation is much higher than the elastic recoil at the moment of the stress cessation (corresponding to the break of lines).

Figure 8 demonstrates the quantitative characteristics of the elasticity of highly concentrated suspensions, which are expressed as the dependence of the elastic modulus on the magnitude of the shear stress. The elastic modulus, *G* was calculated as
(6)$$G=\frac{\sigma }{\gamma_{e}}$$
where *σ* is the shear stress, and *γ**_e_* is the elastic component of the total deformation, which is the difference between the total deformation γ and the value of the plastic component of the deformation *γ**_pl_* (according to Figure 7).

The obtained data indicate that, in general, the elasto-plastic properties of both 75 vol. % and 80 vol. % suspensions do not show any clear dependence on the ratio of coarse and fine fractions in the solid phase, which is especially noticeable in the region of higher shear stresses (>3000 Pa).

The choice of the bimodal composition is based on the theoretical arguments discussed in [27,37,40,41]. An approach for approximation the maximum package limit *φ^M^* for binary systems has been proposed in [40]. The first step in this route was done for a bimodal suspension where the particle size ratio approaches infinity
(7)$$\varphi_{\infty }^{M}=\min\left[\frac{\varphi_{0}^{M}}{1-\xi_{s}},\frac{\varphi_{0}^{M}}{\varphi_{0}^{M}+\left(1-\xi_{L}\right)\left(1-\varphi_{0}^{M}\right)}\right]$$
where $\varphi_{0}^{M}$ is the maximum packing fraction for a monodisperse system; $\xi_{s}=\varphi_{s}/\left(\varphi_{L}+\varphi_{s}\right)$ and $\xi_{L}=\varphi_{L}/\left(\varphi_{L}+\varphi_{s}\right)$ are fractions of small and large particles, respectively.

Then the parameters *f_s_* and *f_L_* as contracting factors were introduced and the assumption that $f_{s}\left(\lambda \right)={\left(1-\lambda^{-1}\right)}^{\alpha }$ and $f_{L}\left(\lambda \right)={\left(1-\lambda^{-1}\right)}^{\beta }$ are found as
(8)$$\xi_{s}^{M}\left(\lambda \right)=1-\frac{f_{s}}{f_{s}+f_{s}\left(1-\varphi_{s}^{M}\right)}\;\mathrm{at}\;1<\lambda \leq \infty$$
where the exponents *α* and *β* are estimated by a fitting procedure.

The final equation that is used for approximating the maximum packing limit *φ^M^* for binary systems with different *λ* and *ξ_s_* was introduced
(9)$$\varphi_{\lambda }^{M}=\min\left[\frac{\varphi_{0}^{M}}{1-f_{s}\xi_{s}},\frac{\varphi_{0}^{M}}{\varphi_{0}^{M}+\left(1-f_{L}\xi_{L}\right)\left(1-\varphi_{0}^{M}\right)}\right]$$where $\varphi_{0}^{M}$—is the maximum packing fraction for a monodisperse system.

The mass fraction of large spheres of the total particle volume is exptressed as $w=\varphi_{L}/\left(\varphi_{L}+\varphi_{s}\right)$, and the final expression for the maximum packing limit [41] is written as:
(10)$$\varphi_{\lambda }^{M}=\min\left[\frac{\varphi_{0}^{M}}{1-w\left(1-\varphi_{0}^{M}\right)},\frac{\varphi_{0}^{M}}{w}\right]$$


According to these theoretical results and using the computer simulation, we have found that the maximum packing limit, in our case, is achieved using particles with λ = 10 and the fraction of large particles equals to 0.75.

Some model formulations based on PEG-400 were prepared with the ratio of the particle size of the available powders close to 10 (λ = 9.4) with the ratio of the fine and coarse phases of 25/75. To verify the possibility of obtaining the composition for powder injection molding with the volume concentration of the disperse phase exceeding 65 vol. %, this mixture of powders was used in preparing a composition containing 75 vol. % of a solid phase.

The rheological properties of this material are presented in Figure 9 and Figure 10.

As seen from Figure 9, these compositions are typical elastic solids.

**Figure 9 polymers-13-02709-f009:**
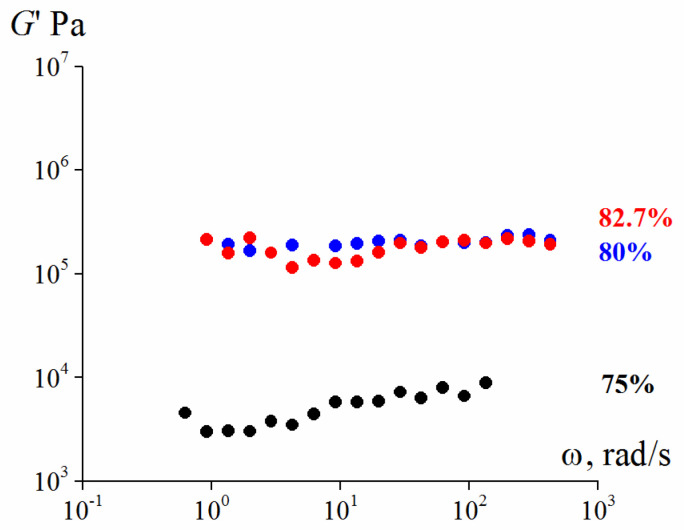
Frequency dependences (at γ = 0.001) of the storage modulus for suspensions with 25% content of smaller particles and λ = 9.4.

Figure 10 presents the values of the shear elastic modulus:

**Figure 10 polymers-13-02709-f010:**
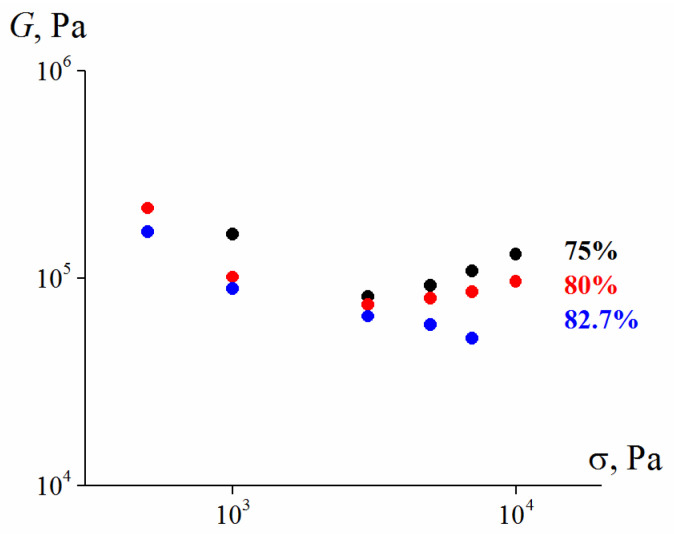
The dependence of the static shear elastic modulus on stress in suspensions with λ = 9.4 and the ratio of large and small particles of the solid phase equals to 25/75.

Comparing two compositions with λ = 2.4 and 9.4, we can conclude that there are no significant discrepancies in the shear elastic modulus values between the compositions with the same content of solid phase.

### 3.2. Compositions for Technological Testing

Two kinds of compositions have been used for evaluating the possibility of using highly concentrated bimodal suspensions in powder injection molding: the first is based on Al bimodal powders, and the second one on aluminum oxide bimodal powders. In both cases, a paraffin-polyethylene matrix as a binder was used. A mixture of aluminum powders of 225 and 24 μm (λ = 9.4) in a ratio of 75 to 25 was used as the solid phase. The total powder content in the composition was 75 vol. %. This value was chosen supposing that a material must have sufficient fluidity to form an article in the injection molding processing.

Figure 11 presents temperature dependences of the apparent viscosity of this composition. It can be seen that temperature suitable for injection molding at relatively low shear rates and moderate shear stress lies in the range of 120–150 °C. Temperatures above this range were not considered, since under strong heating, a significant migration of the plasticizer begins from the composition and paraffin starts to evaporate. So, the temperature range of 120–150 °C was chosen as the most convenient for the molding procedure.

Molecular weight of the matrix used for the injection molding differs significantly from the matrix of the model suspensions. Therefore some elastic recovery of the samples was observed only at a temperature of 100 °C after applying relatively small shear stresses. In addition, as shown in Figure 12, if a rather high load is applied, the so-called spurt occurs with quick and unlimited growth of “apparent” deformations (actually, because of tearing off material from a surface of the instrument and/or wall sliding). The point of the start of the spurt and the magnitude of the corresponding shear stress significantly depend on the temperature.

Measuring the viscoelastic properties of composition shows that a material based on a polymeric matrix demonstrates viscoelastic behavior and *G′′ > G′* over the entire frequency and temperature range (Figure 13). Presence of viscoelastic polymer matrix as a binder causes more noticeable frequency dependences of moduli compared with that for oligomeric PEG.

Figure 14 shows the results of this experiment demonstrating the stability of a sample at a constant shear rate applied over a long period at two different temperatures. The following protocol of these experiments was used. A sample was placed in a gap between two rough plates. Then a constant shear rate was applied, and a change in shear stress over time was recorded every 10 s. Based on these data, the viscosity of the material was calculated. It can be seen that when a higher shear rate is applied, the viscosity of the sample begins to decrease significantly, and to the end of the experiment, the viscosity value decreases by an order of magnitude. In fact, this phenomenon does not correspond to any change in a sample, but sooner with its leakage from the measuring unit. The lower the temperature and the shear rate, the slower this effect occurs, or does not occur at all (Figure 14a, the shear rate is 0.01 s^−1^).

The origin of this effect was cleared up by examination of the samples before and under shearing. The following protocol of the experiment has been applied.

The “before deformation sample” was placed in the gap between the plates of the rheometer and a disk with a diameter of 20 mm and a height of 1.5 mm was formed. After that, the plates were pushed apart and the heating of the device was turned off and the sample cooled to room temperature. Then, the sample was carefully removed from the lower plate and cooled in a liquid nitrogen to obtain a cross-section with smooth edges. The same was done with the “after deformation sample”, only cooling was performed after shearing.

The scanning electron micrographs are shown in Figure 15. Obviously, as a result of deformation over time, a separation of components occurs: the polymer matrix moves towards the boundaries (dark areas near the boundaries of the disk indicate by arrow in Figure 15b), while the solid particles are compacted closer to the center of the sample. This explains the decrease of the apparent “viscosity” due to wall sliding over a less viscous layer, as well as some crowding of samples out the gap between operating surfaces of the instrument during the experiment. This phenomenon should be taken into account in performing injection molding, since too high shear rates during the molding process can cause separation of the material, which in turn can lead to the formation of defects, such as porosity, cavities on the surface and volumetric heterogeneity of the items.

Figure 16 shows samples produced by the injection molding of the above-described composition inside the mold and after their removal from the mold. The injection temperature and pressure were selected taking into account the rheological experimental data. As a result, the formed samples have a high-quality surface without visible defects.

This is so-called “green” parts before sintering and they are demonstrated here only to prove a possibility to prepare articles by the PIM method from very high concentrated suspensions with the content of a solid phase up to 75 vol. %, which is much higher than the standard concentration limit of 65 vol. %.

The rheological approach based on using powder with bimodal particle size distribution developed above for Al-based compositions, we then experienced for the real technological materials taking ceramic aluminum oxide as a typical example. The compositions containing aluminum oxide powders of 6 and 2 mkm in a ratio of 75/25 were prepared. We succeeded to prepare a mixture with the weight fraction of the solid phase of 85 weight %, It should be noted that in usual technological practice, the weight content of ceramics particles does not exceed 78%. So, using the concept of binary mixing, we increased the content of the active component by 7% percent.

Samples in the form of cylinders were obtained by injection molding. After that, the thermal debinding was carried out in the following mode: gradual heating to 350 °C at a rate of 5 degrees per hour. Sintering was performed at a temperature of 1650 °C for 2 h. Figure 17 shows the samples after sintering. They are white and quite smooth without visible surface defects.

The compression strength of these articles was measured using the FP-100/1 Heckert testing machine (traverse speed of 10 mm/min). As a result, the average (for five samples) value of the compression strength was 240 ± 5 MPa. This value corresponds to regular compression strength of engineering ceramics [48].

The apparent porosity of the samples was determined by weighing in air, as well as in water using hydrostatic scales. The apparent porosity value was 10%, which is quite acceptable.

## 4. Conclusions

Based on the analysis of the rheological behavior of model bimodal suspensions and technological samples of highly concentrated compositions, it has been proven that this approach allows for increasing the content of the metallic components in processing by the PIM up to 75 vol. %, which is significantly higher than usually used concentration of 65 vol. %. This possibility is due to retaining plasticity in bimodal suspensions up to higher concentrations, and possibly partly related to the components separation and wall sliding the composition over the viscous layer.

## Figures and Tables

**Figure 1 polymers-13-02709-f001:**
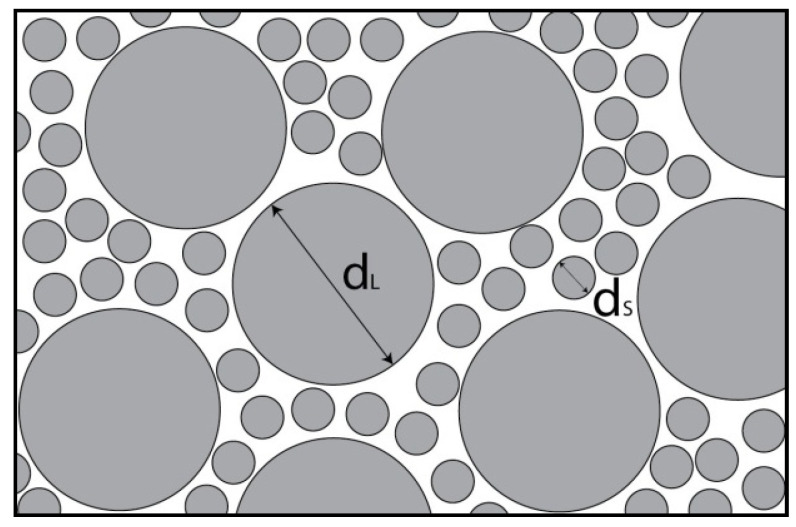
Spatial distribution of spherical particles in a bimodal suspension.

**Figure 2 polymers-13-02709-f002:**
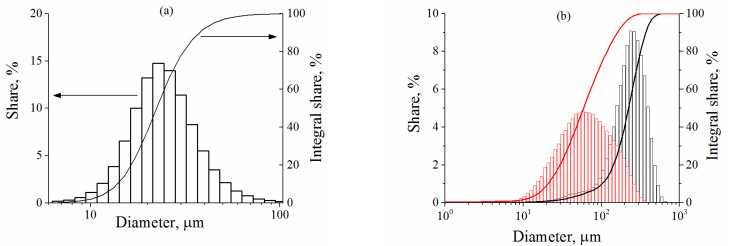
The particle size distribution diagrams of aluminum powders: PAD-1 (**a**) and high-purity aluminum powders (**b**), red lines relate to powder with an average diameter of 58 μm, black ones—225 μm).

**Figure 3 polymers-13-02709-f003:**
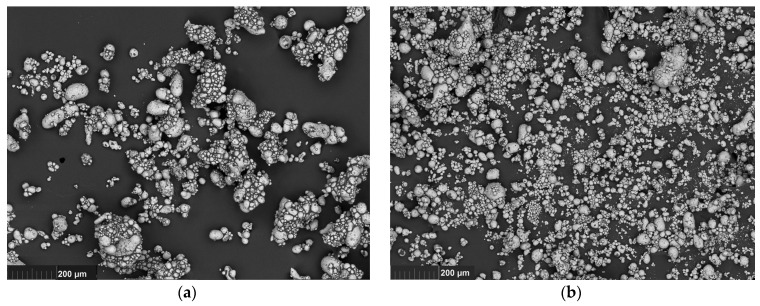
Microphotographs of 80 vol. % suspension with the 20% (**a**) and 40% (**b**) content of smaller particles.

**Figure 4 polymers-13-02709-f004:**
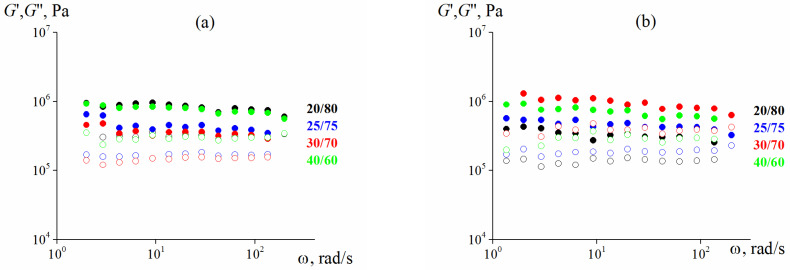
Frequency dependences (at γ = 0.001) of the storage modulus for 75%, (**a**) and 80% suspensions (**b**) with different larger and smaller particles ratios (the storage modulus is marked by dark symbols, the loss modulus—by light ones). In both cases λ = 2.4.

**Figure 5 polymers-13-02709-f005:**
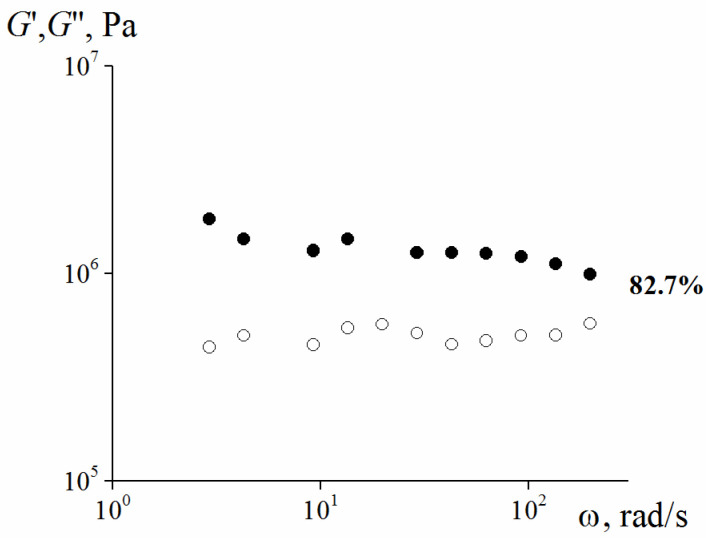
Frequency dependence (at γ = 0.001) of the storage modulus for 82.7 vol. % Scheme 25. content of smaller particles and λ = 2.4 (the storage modulus is marked by dark symbols, the loss modulus—by light ones).

**Figure 6 polymers-13-02709-f006:**
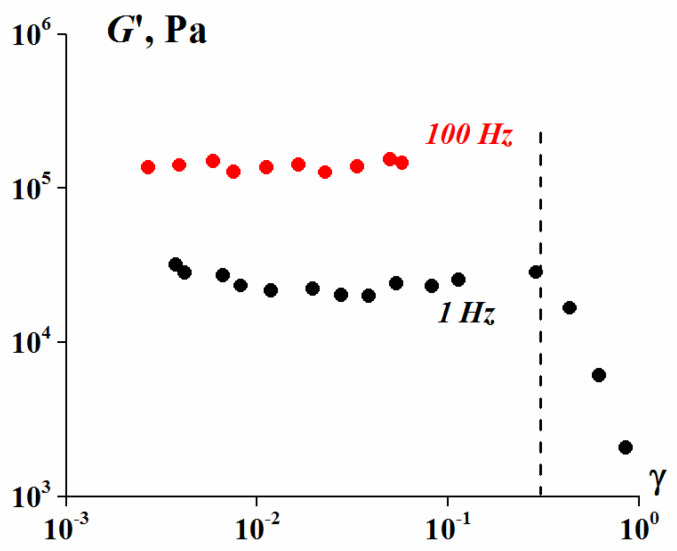
The strain dependence of the storage modulus of the 75 vol. % bimodal suspension (λ = 9.4, 25% content of smaller particles).

**Figure 7 polymers-13-02709-f007:**
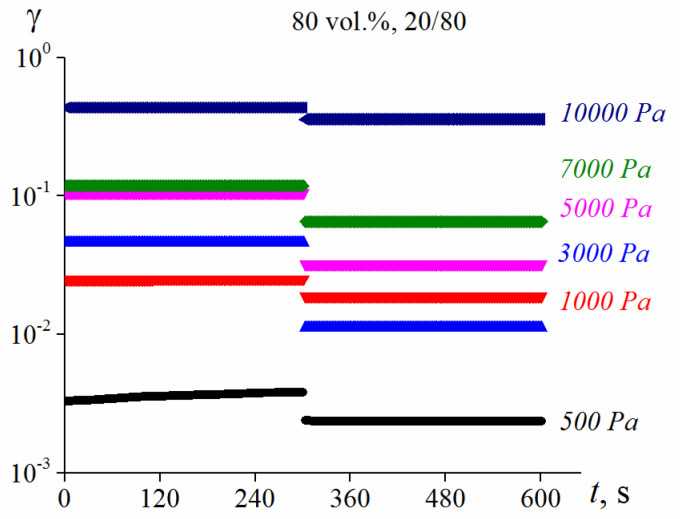
Development of strain and recovery at the applying constant shear stress and its relaxation for the 80 vol. % suspension (λ = 2.4) with a 20% content of smaller particles. Stresses are shown in the Figure.

**Figure 8 polymers-13-02709-f008:**
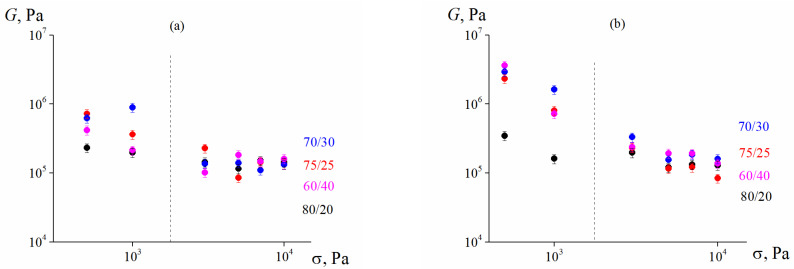
Dependence of the shear elastic modulus on the shear stress in 75 vol. % (**a**) and 80 vol. % (**b**) suspensions (λ = 2.4) with different content of larger and smaller particle ratios.

**Figure 11 polymers-13-02709-f011:**
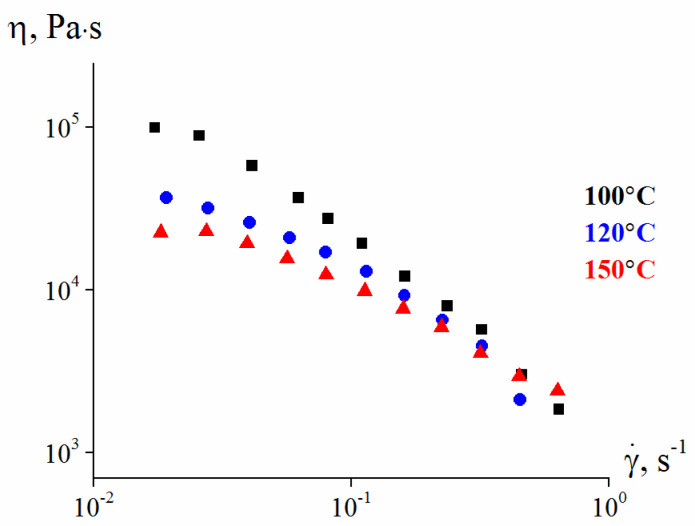
Flow curves of the molding composition at different temperatures.

**Figure 12 polymers-13-02709-f012:**
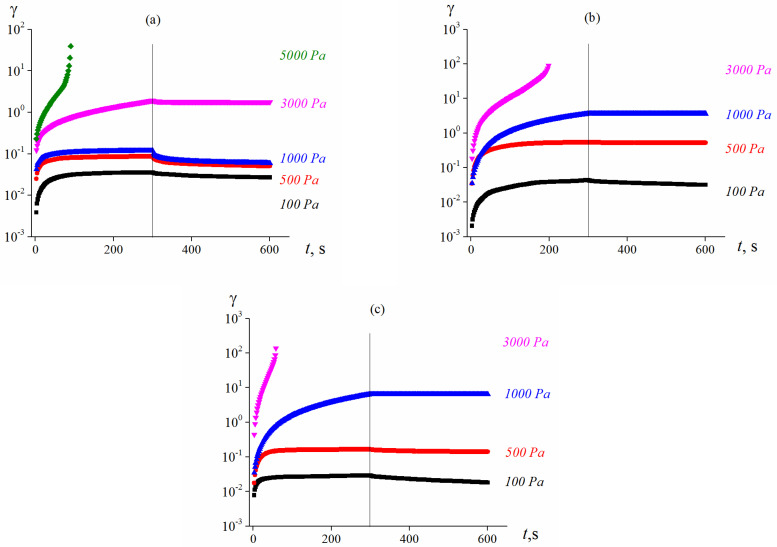
Development and recovery of deformation under constant shear stress and after its cessation at 100 (**a**), 120 (**b**) and 150 °C (**c**) for compositions based on aluminum.

**Figure 13 polymers-13-02709-f013:**
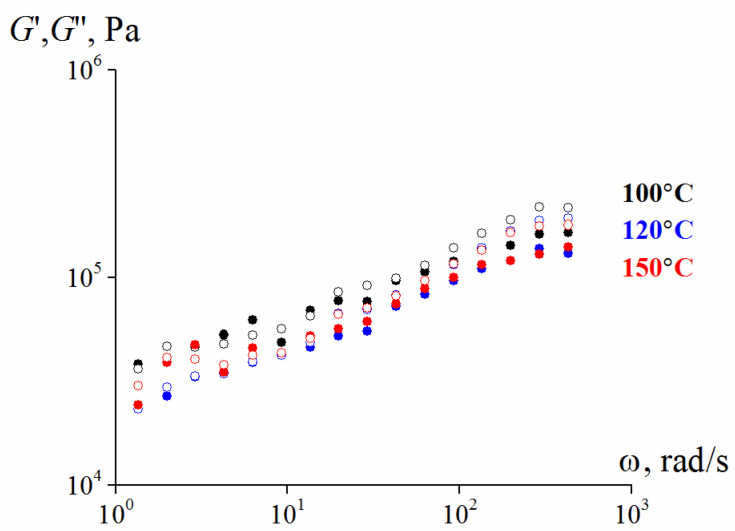
Frequency dependences of the components of dynamic modulus (amplitude 0.001) at different temperatures (the storage modulus is marked by dark symbols, the loss modulus—by light ones).

**Figure 14 polymers-13-02709-f014:**
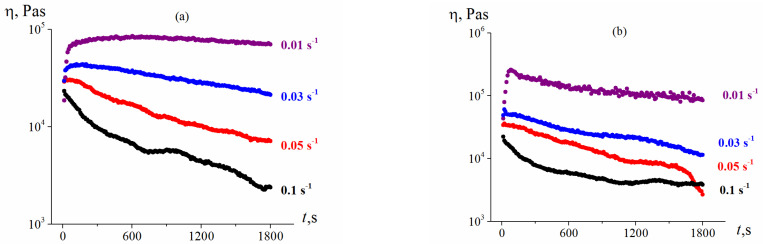
Apparent “viscosity” change over time at long-term observation and the constant shear rate at 120 (**a**) and 150 °C (**b**).

**Figure 15 polymers-13-02709-f015:**
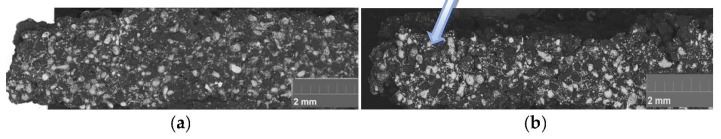
Microphotographs of the cross-sectional split before (**a**) and after (**b**) deformation.

**Figure 16 polymers-13-02709-f016:**
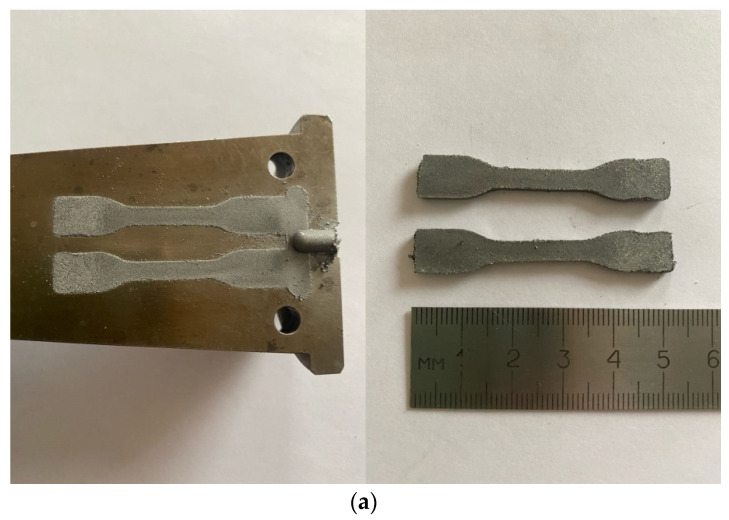
Samples in the form of a standard specimen for mechanical testing (**a**) and a cylinder (**b**), obtained by injection molding.

**Figure 17 polymers-13-02709-f017:**
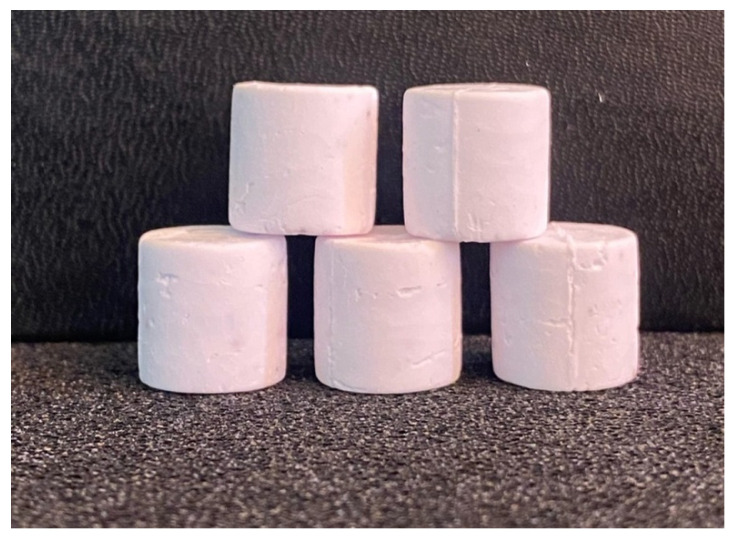
Samples after sintering.

## Data Availability

Not applicable.

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
