# Peer review of "Rheology of Highly Concentrated Suspensions with a Bimodal Size Distribution of Solid Particles for Powder Injection Molding"

_polymers, 2021, doi:10.3390/polym13162709_

Round 1

Reviewer 1 Report

The authors demonstrate that using filler particles of different diameters more filler can be introduced by volume in a suspension as compared to a monodisperse filler. This is intuitively clear. I believe that such an experimental investigation can be important for the PIM technology. However, I have a number of remarks to be addressed before I can recommend the paper for publication.

1) line 205: evidenced --> evident

2) Figures 4 and 5 --> Please incllude the loss modulus G'' to demonstrate that the materials are elastic

3) line 222 --> please quatify "very quickly".

4) Figure 6. I did not understand the discontinuities in the curves. Please explain better.

5) line 238, 239 --> Where is the plastic component of the deformation in Figure 6? Please demonstrate and explain.

6) The passage on lines 240 - 243 is unclear. Where is the value of 3000 Pa in the diagrams?

7) Lines 248, 249 --> Where are these theoretical calculations? Please demonstrate.

8) Does Figure 9 shows the static shear modulus?

9) At which temeperature the data in Figures 4 - 9 is obtained? At room temperature?

10) I believe that Figure 15 a does not show a rod.

11) Line 22 --> Please define and quantify which mechanical properties are "quite acceptable".

12) My major objection is that the authors claimed to have theoretical arguments (line 14) but does not have a theoretical part in their paper.

Author Response

We are grateful to both Reviewers for their attention to our work and valuable comments. All of them are taken into account and the necessary changes were made along with the revised manuscript.

Dear Reviewer 1,

1) line 205: evidenced --> evident

            Sorry, corrected

2) Figures 4 and 5 --> Please include the loss modulus G'' to demonstrate that the materials are elastic

   Experimental data were added.    

3) line 222 --> please quatify "very quickly".

            Corrected: faster than 1 s.

4) Figure 6. I did not understand the discontinuities in the curves. Please explain better.

The following explanation was added:

The break on the curves corresponds to the cessation of stress and elastic recoil.

5) line 238, 239 --> Where is the plastic component of the deformation in Figure 6? Please demonstrate and explain.

The following explanation was added:

Indeed, deformation at the right parts of this Figure having the sense of irreversible part of the full deformation is much higher than the elastic recoil at the moment of the stress cessation (corresponding to the break of lines).

6) The passage on lines 240 - 243 is unclear. Where is the value of 3000 Pa in the diagrams?

  This value is shown by dotted lines in modified calculations.

7) Lines 248, 249 --> Where are these theoretical calculations? Please demonstrate.

Yes, we added the Reference where these calculations have been made and explained the theoretical background of calculations.

The following paragraph was added in text:

The choice of the bimodal composition is based on the theoretical arguments discussed in [27, 37, 40, 41]. An approach for approximation the maximum package limit φM for binary systems has been proposed in [40]. The first step in this route was done for a bimodal suspension where the particle size ratio approaches infinity:

where   - is the maximum packing fraction for a monodisperse system;    and  are fractions of small and large particles, respectively.

Then the parameters fs and fL as contracting factors were introduced and the assumption that   and  are found as

  at

where the exponents α and β are estimated by a fitting procedure.

The final equation that is used for approximating the maximum packing limit φM for binary systems with different λ and ξwas introduced:

where  - is the maximum packing fraction for a monodisperse system.

The mass fraction of large spheres of the total particle volume is exptressed as  and the final expression for the maximum packing limit [41] is written as

According to these theoretrical results and using the computer simulation, we have found that the maximum packing limit in our case, is achieved using particles with λ=10 and the fraction of large particles equals to 0.75.

8) Does Figure 9 shows the static shear modulus?

Yes, this definition was added in the capture.

9) At which temperature the data in Figures 4 - 9 is obtained? At room temperature?

Yes. The whole body of measurements were made at room temperature. This comment was added to Experimental part.

10) I believe that Figure 15 a does not show a rod.

Surely, this is a spade–standard specimen for mechanical testing.

11) Line 22 --> Please define and quantify which mechanical properties are "quite acceptable".

Well, we corrected this for "of the usual corundum articles".

12) My major objection is that the authors claimed to have theoretical arguments (line 14) but does not have a theoretical part in their paper.

Well, we agree with this comment and included corresponding explanations in text (pages 10-11).

Reviewer 2 Report

In the paper, an extensive study on rheology of the highly concentrated suspensions of Al with low-molecular-weight poly(ethylene glycol) as a binder (with a bimodal size distribution of solid particles) for Powder Injection Molding has been performed.

The paper is interesting and valuable both from a scientific and practical point of view. In general, it is clearly written and well documented.

However, there are some comments:

  • PIM technique should be briefly described due to the wide range of readers, and not only specialists.
  • Injection Molding experiment is not clearly described. There is lack of some data to analyse the results, e.g. injection rate and samples dimensions (Figure) to evaluate shear rate.
  • The experiment was performed at very low pressure (6 bar), however the viscosity was relatively high (Figure 10, at very low shear rate), etc.

Author Response

We are grateful to both Reviewers for their attention to our work and valuable comments. All of them are taken into account and the necessary changes were made along with the revised manuscript.

Dear Reviewer 2,

  • PIM technique should be briefly described due to the wide range of readers, and not only specialists.

Thank you for this comment. The following text has been added:

This technology is a combination of two areas of manufacturing: polymer injection molding and powder technology. There are four basic stages: preparing a feedstock, molding, debinding, and sintering.  Polymer binder and metal or ceramic powder are two main components of the feedstock. Polymers in combination with some additives provide a possibility of processing the various parts from feedstocks. However, all components except the powder should be removed after injection molding in order to obtain a metal or ceramic item. This stage is called debinding. The debinding stage can have different versions such as thermolysis or removing by dissoilution, or catalytic debinding. The final stage is sintering when powder particles are sintered with each other. 

  • Injection Molding experiment is not clearly described. There is lack of some data to analyse the results, e.g. injection rate and samples dimensions (Figure) to evaluate shear rate.
  • The experiment was performed at very low pressure (6 bar), however the viscosity was relatively high (Figure 10, at very low shear rate), etc.

The following injection molding mode was used:

Shot volume (V) - 6 cm3, injection time (t) 4 s,  barrel radius 6 mm, nozzle radius is 2 mm, injection pressure 6 bar, melt temperature 150°C.

Based on these parameters, volumetric flow rate (Q) was calculated:

The shear rate on the barrel wall ( )  and nozzle wall ( )  was calculated, as well:

Round 2

Reviewer 1 Report

The authors have reacted properly to the reviewers' comments.

The paper can be recommended for publication.